# Prognostic Value of Mid-Region Proadrenomedullin and In Vitro Interferon Gamma Production for In-Hospital Mortality in Patients with COVID-19 Pneumonia and Respiratory Failure: An Observational Prospective Study

**DOI:** 10.3390/v14081683

**Published:** 2022-07-30

**Authors:** Davide Mangioni, Massimo Oggioni, Liliane Chatenoud, Arianna Liparoti, Sara Uceda Renteria, Laura Alagna, Simona Biscarini, Matteo Bolis, Adriana Di Modugno, Marco Mussa, Giulia Renisi, Riccardo Ungaro, Antonio Muscatello, Andrea Gori, Ferruccio Ceriotti, Alessandra Bandera

**Affiliations:** 1Infectious Diseases Unit, Foundation IRCCS Ca’ Granda Ospedale Maggiore Policlinico, 20122 Milan, Italy; arianna.liparoti@unimi.it (A.L.); laura.alagna@policlinico.mi.it (L.A.); simona.biscarini@policlinico.mi.it (S.B.); matteo.bolis@policlinico.mi.it (M.B.); marco.mussa@policlinico.mi.it (M.M.); giulia.renisi@policlinico.mi.it (G.R.); riccardo.ungaro@policlinico.mi.it (R.U.); antonio.muscatello@policlinico.mi.it (A.M.); andrea.gori@unimi.it (A.G.); alessandra.bandera@unimi.it (A.B.); 2Department of Pathophysiology and Transplantation, University of Milano, 20122 Milan, Italy; 3Clinical Laboratory, Foundation IRCCS Ca’ Granda Ospedale Maggiore Policlinico, 20122 Milan, Italy; massimo.oggioni@policlinico.mi.it (M.O.); sara.ucedarenteria@policlinico.mi.it (S.U.R.); adriana.dimodugno@policlinico.mi.it (A.D.M.); ferruccio.ceriotti@policlinico.mi.it (F.C.); 4Laboratory of Clinical Epidemiology, Department of Public Health, Istituto di Ricerche Farmacologiche Mario Negri IRCCS, 20156 Milan, Italy; liliane.chatenoud@marionegri.it

**Keywords:** mid-region proadrenomedullin, interferon gamma, MR-proADM, quantiferon monitor, SARS-CoV-2

## Abstract

Coagulopathy and immune dysregulation have been identified as important causes of adverse outcomes in coronavirus disease (COVID-19). Mid-region proadrenomedullin (MR-proADM) is associated with endothelial damage and has recently been proposed as a prognostic factor in COVID-19. In non-COVID-19 immunocompromised patients, low in vitro interferon gamma (IFNγ) production correlates with infection risk and mortality. This prospective, monocentric, observational study included adult patients consecutively admitted with radiologic evidence of COVID-19 pneumonia and respiratory failure. MR-proADM and in vitro IFNγ production were measured at T0 (day 1 from admission) and T1 (day 7 from enrollment). One hundred patients were enrolled. Thirty-six percent were females, median age 65 (Q1–Q3 54.5–75) years, and 58% had ≥1 comorbidity. Only 16 patients had received COVID-19 vaccination before hospitalization. At admission, the median PaO2:FiO2 ratio was 241 (157–309) mmHg. In-hospital mortality was 13%. MR-proADM levels differed significantly between deceased and survivors both at T0 (1.41 (1.12–1.77) nmol/L vs. 0.79 (0.63–1.03) nmol/L, *p* < 0.001) and T1 (1.67 (1.08–1.96) nmol/L vs. 0.66 (0.53–0.95) nmol/L, *p* < 0.001). In vitro IFNγ production at T0 and T1 did not vary between groups. When only the subset of non-vaccinated patients was considered, both biomarkers at T1 resulted significantly associated with in-hospital mortality. AUROC for MR-proADM at T0 to predict in-hospital mortality was 0.87 (95%CI 0.79–0.94), with the best cut-off point at 1.04 nmol/L (92% sensitivity, 75% specificity and 98% negative predictive value). In patients with COVID-19 pneumonia and different degrees of respiratory failure, MR-proADM at admission and during hospitalization resulted strongly associated with in-hospital mortality. Low in vitro IFNγ production after the first week of hospitalization was associated with mortality in non-vaccinated patients possibly identifying the subgroup characterized by a higher degree of immune suppression.

## 1. Introduction

Symptomatic SARS-CoV2 infection (coronavirus disease, COVID-19) is characterized by a wide spectrum of disease severity. Moderate disease is reported in about 80% of cases while severe forms requiring hospitalization occur in the remaining 20%, with 5% requiring intensive care unit (ICU) admission [1]. Several clinical-epidemiological variables (i.e., gender, age, comorbidities) and laboratory abnormalities during hospitalization (i.e., lymphopenia, increase in markers of inflammation or coagulopathy) have been associated with worse outcomes [2,3] Currently, however, no biological marker has proved good reliability in predicting patients’ outcome (i.e., ICU admission/death).

At least two major pathobiological mechanisms have been recognized in the pathogenesis of acute respiratory distress syndrome (ARDS) associated with COVID-19, namely: (i) a deregulated immune response with an increase in proinflammatory cytokines and a concomitant inhibition and functional exhaustion of antiviral lymphocytes [4,5,6] and (ii) an endothelial dysfunction and hypercoagulable state associated with thrombotic microangiopathy and endothelitis [7,8]. Interferon γ (IFNγ) is a major product of Th1-mediated immune response and orchestrates Th1 effector mechanisms, as further activation of innate immunity. Outside COVID-19, its measurement has been recently proposed as a functional marker of the immune status in particular settings of immunosuppressed patients (liver cirrhosis, solid organ transplant and bone marrow transplant). Clinical studies have shown significant differences in IFNγ production between healthy and immunosuppressed subjects [9]. Low in vitro IFNγ production has also been associated with an increased risk of infections in fragile patients [9,10,11]. In COVID-19, while impaired type I IFN (i.e., IFNα and IFNβ) activity has been observed in patients with a severe disease course [6], in vitro IFNγ production has been less studied. IFNγ levels obtained at hospital admission appear to inversely correlate with disease severity and predict the risk of complications such as ICU access, deep vein thrombosis, secondary bacterial infections, organ failure and death [5,12]. Adrenomedullin (ADM) is a vasoactive peptide that regulates endothelial function and microcirculation. It has various biological effects as a vasodilator, positive inotropic, diuretic and bronchodilator. Elevated levels of ADM representing endothelial damage and microvascular alteration have been found in patients with sepsis and organ dysfunctions such as heart and respiratory failure [13]. The mid-regional fragment of pro-adrenomedullin (MR-proADM) acts as an excellent surrogate of ADM levels, and its longer half-life enables its use as an early indicator of organ dysfunction, adverse evolution and mortality in sepsis/septic shock and other infections [14,15]. Recently, this biomarker has gained interest in COVID-19 patients and proved to correlate with disease severity and mortality in studies both in and outside the intensive care unit [16,17,18,19], with only a few analyzing its prognostic value over time [17,19,20].

The aim of this study was to evaluate MR-proADM and in vitro IFNγ production as prognostic markers of mortality in patients with COVID-19 pneumonia and respiratory failure at hospitalization.

## 2. Methods

This is an observational, prospective, single-center study involving adult patients with SARS-CoV-2 infection consecutively admitted to the FoundationIRCCS Ca’ Granda Ospedale Maggiore Policlinico in Milan, Italy, from 23 February to 26 October 2021. At the time of enrolment, all patients had radiological evidence of COVID-19 pneumonia and respiratory failure at different degrees with peripheral oxygen saturation (SpO2) in room air <94% and/or need for supplemental oxygen. Low-flow oxygen support was defined as nasal cannula or venturi mask; high-flow oxygen devices included high-flow nasal cannula and continuous positive airway pressure (CPAP). Exclusion criteria were: expected survival <48 h, active hematological or autoimmune diseases, HIV/AIDS, solid organ transplantation or any use of immunosuppressive drugs including chronic high-dose corticosteroids (prednisone ≥20 mg/day or equivalent for ≥4 weeks). For enrolled patients, blood samples were collected at T0 (enrollment, i.e., ≤48 h from admittance to the emergency department) and T1 (i.e., on day 7 from enrollment). Patients were managed as per clinical practice during hospital stay, according to local and national guidelines in use at the time of enrollment (*Italian Society of Infectious Diseases and Tropical Medicine*—SIMIT, edition 3.0, November 2020). Standard treatment of COVID-19 pneumonia consisted of dexamethasone 6 mg daily for 10 days, with the addition of remdesivir if low oxygen support was required. In case of severe ARDS or rapid worsening of respiratory failure, steroid dosage was increased to methylprednisolone 1 mg/kg/die for at least 7 days with subsequent tapering down according to clinical response.

The study was approved by the institutional review board (Ethic Committee Milano Area 2, #176_2021) and was conducted in accordance with the Helsinki Declaration. Written informed consent was obtained before enrollment.

### 2.1. Mid-Regional Proadrenomedullin (MR-proADM)

For MR-proADM evaluation, 5 mL of whole blood was obtained by venipuncture and collected in EDTA tubes. Tubes were centrifuged for 15 min at 3000× *g*, and MR-proADM plasmatic levels were determined with B.R.A.H.M.S MR-proADM KRYPTOR^®^ assay (KRYPTOR™, B.R.A.H.M.S Thermo Fisher Scientific, Dreieich, Germany) by an automated B.R.A.H.M.S KRYPTOR^®^ analyzer using Time-Resolved Amplified Cryptate Emission (TRACE) technology. The Immunoassay has a limit of detection (LOD) of 0.05 nmol/L and a functional assay sensitivity of 0.25 nmol/L.

### 2.2. In Vitro Interferon Gamma (IFNγ) Production

In vitro IFNγ production was measured with QuantiFERON Monitor^®^ (QFM^®^) assay (QIAGEN, Hilden, Germany). The analysis was performed according to the manufacturer’s instructions. One mL of peripheral blood was collected in lithium heparin tubes, kept at room temperature and transferred within 8 h from collection to the QFM assay tube (QFM Blood Collection Tubes) for stimulation with lyophilized spheres (QFM LyoSphere^TM^) containing anti-CD3 and R R848 immune ligands that stimulate T-cell receptor (TCR) and Toll-like receptor 7/8 (TLR 7/8), respectively. The stimulated blood samples were transferred to an incubator at 37 °C for 16 to 24 h, and then plasma was collected after centrifugation for 15 min at 3000× *g*. Plasmatic levels of IFNγ were then measured by QFM enzyme-linked immunosorbent assay (QFM ELISA) according to the manufacturer’s instructions and reported in IU/mL. The detection limit of the QFM ELISA is 0.065 IU/mL. QuantiFERON Monitor^®^ Analysis Software v4.00.1 (QIAGEN, Germany) was used to analyze raw data and calculate the results.

### 2.3. Statistical Analysis

Continuous variables are presented as median (interquartile range (IQR)), categorical variables as frequency (percentage), for all patients and stratified by vital status at discharge. Comparisons between groups were performed with independent sample *t*-test, Mann–Whitney U test, chi-squared test or Fisher exact test, depending on variable distribution. The relation of MR-proADM and in vitro IFNγ production with the outcome was further analyzed through univariate and multivariable exact logistic regression models (in order to address issues of separability), run to take into consideration confounders. As potential confounders, we considered the main patients’ characteristics at admission (see Table 1), found to be statistically significant (*p* < 0.05) in the univariate analysis. However, if Pearson or Spearman correlation coefficient (according to variable distribution) was ≥0.30, the variable with the lowest *p* value was retained in the model. Given the low sample size, the models could include at most one confounding variable. Therefore, instead of showing the results of the age and PaO2:FiO2 ratio-adjusted models (i.e., the two chosen covariates representing patients’ frailty and severity of the disease, respectively), we constructed a severity score (estimated through an exact logistic regression model predicting mortality based on age and PaO2:FiO2) and adjusted models by this severity score. For this same reason, in the subgroup analysis of non-vaccinated patients (6 deceased in total), only the univariate analysis was run. Exact odds ratios (ORs) and corresponding 95% confidence intervals (95%CIs) were reported as a measure of association. Since in vitro IFNγ production had a very skewed distribution toward high values, we also analyzed its logarithmic transformation (base e), and we reported the *p* values of the analyses corresponding to the latter. C-statistics with area under the operating receiver curve (AUROC) were reported as measure of discrimination, and optimal cut-off points of MR-proADM at T0 and T1 were assessed by the Youden rule (Youden, 1950). All tests were two-tailed, and a *p*-value <0.05 was considered statistically significant. All analyses were performed in SAS Version 9.4 (SAS Institute Inc., Cary, NC, USA).

## 3. Results

### 3.1. Study Population and Comparison between Survivors and Deceased

One hundred patients were enrolled from 23 February to 26 October 2021. Median age was 65 (54.5–75) years, 36 were women, and the median body mass index (BMI) was 25.8 (23.7–29.7) kg/m^2^. Anamnestic, clinical and laboratory characteristics of the study population at enrollment (T0) and after 7 days (T1) are reported in Table 1 and Appendix A.

**Table 1 viruses-14-01683-t001:** Clinical and laboratory characteristics at enrollment (T0) and after 7 days (T1), overall and for survivors and deceased.

	Overall (*n* = 100)	Survivors (*n* = 87)	Deceased (*n* = 13)	*p* Value
**Patient Characteristics**				
Age	65 (54–75)	63 (53–73)	77 (73–82)	**0.003**
Gender female	36 (36.0)	31 (35.6)	5 (38.5)	0.843
BMI	25.8 (23.7–29.7)	25.5 (23.4–29.4)	29.3 (27–31.9)	0.066
CCI	0 (0–1)	0 (0–1)	1 (1–3)	**<0.001**
Hypertension	43 (43.0)	35 (40.2)	8 (61.5)	0.148
Ever smoker	19 (19.0)	17 (19.5)	2 (15.4)	1.00
Any comorbidity #	58 (58.0)	46 (52.9)	12 (92.3)	**0.007**
Days from symptom onset to hospitalization	7 (5–10)	7 (5–10)	8.5 (4.5–10)	0.604
At least one dose of COVID-19 vaccine before admission				**0.006**
No	74 (74.0)	67 (77.0)	7 (53.9)	
Yes	16 (16.0)	15 (17.2)	1 (7.7)	
Missing	10 (10.0)	5 (5.8)	5 (38.5)	
**Clinical and Laboratory Characteristics at T0**				
Steroid intake before admission	38 (38.0)	36 (41.9)	2 (15.4)	0.123
Body temperature, °C	36.5 (36.1–37.5)	36.5 (36.0–37.5)	37.0 (36.2–38.0)	0.254
PaO2:FiO2 ratio	241 (157–309)	248 (167–314)	150 (111–247)	**0.023**
Respiratory support ^				**0.026**
None/low-flow oxygen	64 (64.0)	60 (69.0)	4 (30.8)	
Non-invasive ventilation	31 (31.0)	23 (26.4)	8 (61.5)	
Mechanical ventilation	5 (5.0)	4 (4.6)	1 (7.7)	
NIH ordinal scale †				0.051
4	2 (2.0)	2 (2.3)	0 (0.0)	
5	62 (62.0)	58 (66.7)	4 (30.8)	
6	31 (31.0)	23 (26.4)	8 (61.5)	
7	5 (5.0)	4 (4.6)	1 (7.7)	
Steroid intake §				**0.001**
Standard dose	79 (79.0)	73 (84.9)	6 (46.1)	
High dose	20 (20.0)	13 (15.1)	7 (43.9)	
C-reactive protein, mg/dL	7.1 (4.4–12.1)	7.3 (3.6–12.4)	7.0 (5.1–8.5)	0.918
Lymphocyte, cell/µL	860 (600–1290)	900 (600–1300)	700 (500–1200)	0.118
Creatinine, mg/dL	0.9 (0.8–1.1)	0.9 (0.8–1.1)	1.0 (0.9–1.2)	0.386
D dimer, µg/L (*n* = 84)	770 (559–1335)	733 (537–1278)	1116 (841–1543)	0.077
**Clinical and Laboratory Characteristics at T1**				
PaO2:FiO2 ratio (*n* = 48)	215 (143–260)	223 (194–284)	113 (98–170)	**0.001**
Respiratory support ^				**0.001**
None	23 (28.4)	23 (33.3)	0 (0.0)	
Low-flow oxygen	14 (17.3)	14 (20.3)	0 (0.0)	
Non-invasive ventilation	40 (49.4)	30 (43.5)	10 (83.3)	
Mechanical ventilation	4 (4.9)	2 (2.9)	2 (16.7)	
NIH ordinal scale †				**0.003**
3	9 (11.1)	9 (13.0)	0 (0.0)	
4	14 (17.3)	14 (20.3)	0 (0.0)	
5	14 (17.3)	14 (20.3)	0 (0.0)	
6	40 (49.4)	30 (43.5)	10 (83.3)	
7	4 (4.9)	2 (2.9)	2 (16.7)	
Steroid intake §				0.057
No steroid	2 (2.5)	2 (2.9)	0 (0.0)	
Standard dose	47 (58.0)	43 (61.8)	4 (33.3)	
High dose	32 (39.5)	24 (35.3)	8 (66.7)	
C-reactive protein, mg/dL	1.0 (0.4–2.8)	0.8 (0.4–2.1)	5.4 (0.9–12)	**0.013**
Lymphocyte, cell/µL	1290 (820–1920)	1400 (1100–2000)	600 (400–1000)	**<0.001**
Creatinine, mg/dL	0.8 (0.7–0.9)	0.8 (0.7–0.9)	0.8 (0.8–1.0)	0.442
D dimer, µg/L (*n* = 58)	1005 (624–1980)	898 (541–1785)	1693 (1105–4073)	**0.028**
Days of hospitalization	12.5 (8.5–21.0)	12.0 (8.0–20.0)	20 (11.0–25.0)	0.063

Categorical variables are expressed as counts and percentages and continuous variables as medians and interquartile ranges. *p* values < 0.05 are reported in bold. Legend: BMI, body mass index; CCI, Charlson Comorbidity Index; # details are reported in Appendix A; ^ low-flow oxygen (nasal cannula, venturi mask), non-invasive ventilation (high-flow nasal cannula, continuous positive airway pressure (CPAP)); † 3: hospitalized, not requiring supplemental oxygen and no longer requiring ongoing medical care (used if hospitalization was extended for infection-control or other nonmedical reasons), 4: hospitalized, not requiring supplemental oxygen but requiring ongoing medical care (related to COVID-19 or to other medical conditions), 5: hospitalized, requiring any supplemental oxygen, 6: hospitalized, requiring noninvasive ventilation or use of high-flow oxygen devices, 7: hospitalized, receiving invasive mechanical ventilation or extracorporeal membrane oxygenation (ECMO); § standard dose if dexamethasone or methylprednisolone <1 mg/kg/die, high dose if methylprednisolone >1 mg/kg/die or equivalent.

Patients’ frailty was moderately low on average, with Charlson Comorbidity Index (CCI) ≥3 only in 10 patients. Median time from symptom onset to hospitalization was 7 (5–10) days. Thirty-eight patients received steroid treatment before hospitalization. Only 16 patients, all admitted starting from May 2021, had received a COVID-19 vaccine at the time of hospitalization (5 with the full vaccination course). This is consistent with the vaccination campaign in Italy, which was implemented for elderly and fragile patients from March 2021. At enrollment (T0), the median PaO2:FiO2 ratio was 241 (157–309), with 64 patients on room air or low-flow oxygen, 31 on high-flow oxygen support and 5 mechanically ventilated. All patients but one received steroid therapy, 79 with standard dose and 20 with high-dose steroid. C-reactive protein (CRP) was moderately elevated with a median value of 7.1 (4.4–12.1) mg/dL, and the median lymphocyte count was 860 (600–1290)/mmc. After 7 days from enrollment (T1), 81 patients were still hospitalized, 18 were discharged, and 1 died. Median PaO2:FiO2 ratio reduced to 215 (143–260), with 37/81 patients (45.7%) on room air or low-flow oxygen, 40/81 (49.4%) on high-flow oxygen support and 4/81 (4.9%) mechanically ventilated. Seventy-nine patients were still receiving steroid therapy, 47/79 (59.5%) with standard dose and 32 (40.5%) with high dose. Compared to T0, inflammatory markers globally improved, with a reduction in CRP to 1 (0.4–2.8) mg/dL and recovery of lymphocyte count at 1290 (820–1920)/mmc. Median length of hospitalization was 12.5 (8.5–21.0) days. During the hospitalization, 17 patients had at least one episode of secondary infection, 5 had thromboembolic events, and 4 patients had both complications (Appendix A).

Of the 100 patients enrolled, 13 died and 87 survived. Deceased patients were significantly older than survivors with a median age of 77 (73–82) years compared to 63 (53–73) years (*p* = 0.003). One or more comorbidities were present in 12/13 (92%) and 46/87 (53%) of patients, respectively (*p* = 0.007). At T0, the median PaO2:FiO2 ratio differed between groups with 150 (111–247) in deceased and 248 (167–314) in survivors (*p* = 0.023). Similarly, invasive/non-invasive mechanical ventilation as respiratory support at T0 was more frequent in deceased compared to survivors (9/13 (69.2%) and 27/87 (31%), *p* = 0.026), as was the use of high-dose steroid therapy (7/13 (43.9%) and 13/86 (15.1%), *p* = 0.001). Inflammatory markers differed between deceased and survivors at T1 but not at T0, with lymphocyte count 600 (400–1000)/mmc compared to 1400 (1100–2000)/mmc (*p* < 0.001), CRP 5.4 (0.9–12) mg/dL compared to 0.8 (0.4–2.1) mg/dL (*p* = 0.013) and d dimer levels 1693 (1105–4073)/mmc compared to 898 (541–1785)/mmc (*p* = 0.028), respectively (Table 1).

### 3.2. Association of MR-proADM and IFNγ Production Levels with In-Hospital Mortality

MR-proADM and in vitro IFNγ production were assessed in all patients at T0 and T1. No association was found between MR-proADM and IFNγ (Spearman correlation coefficient 0.18). Table 2 shows median values of MR-proADM, in vitro IFNγ production and its logarithm (Log IFNγ) of the entire study population and in the subset of non-vaccinated patients, comparing deceased to survivors.

At T0, medians of MR-proADM, IFNγ production and Log IFNγ were 0.84 (0.66–1.2) nmol/L, 4.50 (0.85–17.60) IU/mL and 1.50 (−0.16–2.87), respectively. MR-proADM levels differed significantly between deceased and survivors both at T0 (1.41 (1.12–1.77) nmol/L versus 0.79 (0.63–1.03) nmol/L, *p* < 0.001) and T1 (1.67 (1.08–1.96) nmol/L versus 0.66 (0.53–0.95) nmol/L, *p* < 0.001). In vitro IFNγ production at T0 did not differ between groups, while at T1, changes occurred although not reaching statistical significance, with lower median values in deceased compared to survivors (Log IFNγ: 0.2 (−0.5–1.1) versus 1.8 (−0.3–3. 0), *p* = 0.083). Interestingly, when subgroup analysis was conducted on unvaccinated patients, both biomarkers at T1 resulted significantly associated with in-hospital mortality (Table 2). Yet, caution should be paid when interpreting these results in light of the number of missing data on vaccination status, particularly for deceased patients (5/13, 38%), which further reduces the sample size under study (Table 1). All results remained comparable even after accounting for the potential confounding effect of severity score (based on age and PaO2:FiO2 ratio at T0) in multivariable exact logistic regression models.

### 3.3. Prognostic Values of MR-proADM and In Vivo IFNγ Production

Figure 1 shows the prognostic performances of MR-proADM and in vitro IFNγ production for in-hospital mortality censored at 30 days.

At T0, MR-proADM displayed good prognostic performance with AUROC 0.87 (95%CI 0.79–0.94), while IFNγ production showed only suboptimal performance (AUROC Log IFNγ 0.53, 95%CI 0.37–0.70) (Figure 1A). When measured at T0, MR-proADM at the cut-off of 1.04 nmol/L was characterized by 92.3% sensitivity, 75% specificity and a 98% negative predictive value for mortality. At T1, the prognostic capacity of both biomarkers improved, although the performance of IFNγ production remained quite low, with AUROC of MR-proADM and Log IFNγ at 0.91 (95%CI 0.84–0.98) and 0.67 (CI95% 0.50–0.90), respectively (Figure 1B). 

When considering only the 74 non-vaccinated patients, the prognostic performance of both biomarkers increased, especially at T1, up to good accuracy for MR-proADM (AUROC 0.94; 95%CI 0.86–1.00) and moderate accuracy for Log IFNγ (AUROC 0.74; 95%CI 0.59–0.89) (Figure 2).

The prognostic performance of other inflammatory markers (lymphocyte count, CRP, ferritin levels) is described in Appendix A, both at T0 (panel A) and T1 (panel B). None exhibit significant prognostic value, with the lower bounds of the 95% confidence intervals never exceeding 70% with the sole exception of lymphocyte count at T1. Results remained comparable when only non-vaccinated patients were considered (data not shown)

## 4. Discussion

In this prospective, monocentric study conducted on consecutively enrolled adult patients with COVID-19 pneumonia and different degrees of respiratory failure, we observed a strong correlation between MR-proADM levels both at admission and after the first week of hospitalization and in-hospital mortality. Low in vitro IFNγ production after the first week of hospitalization (but not at admission) was also associated with in-hospital mortality in non-vaccinated patients.

Early identification of patients at high risk for severe disease or death has become crucial in COVID-19 management. Currently, the decision to initiate immunomodulatory drugs is guided by the clinical worsening with increasing oxygen need and systemic inflammation markers on a case-by-case basis [21]. Since the beginning of the pandemic, a number of biological markers have been identified to stratify COVID-19 patients at risk of clinical complications and mortality. Among these, the most widely used are CRP and lymphocyte count as indicators of immune system dysregulation and d-dimer and fibrinogen as markers of coagulopathy [2,3]. Yet, in the current pandemic scenario, earlier and more accurate predictors of adverse clinical outcomes in COVID-19 patients are still to be defined and highly warranted.

Severe forms of COVID-19 are mainly driven by two pathobiological mechanisms: immune system dysregulation and coagulopathy. SARS-CoV-2 infects both pulmonary alveolar cells and alveolar macrophages through the binding of ACE-2 protein, triggering the release of numerous pro-inflammatory cytokines known as a “cytokine storm”. This amplifies the damaging process by increasing endothelial dysfunction and vasodilation of pulmonary vessels [4]. Moreover, infected patients have concomitant inhibition and functional depletion of antiviral lympho-monocytes, which is found proportionate to the clinical severity of the disease [4,5,6]. Endothelial dysfunction is either immune-mediated or a direct consequence of viral infection and is the principal determinant of microvascular alterations and coagulopathy [7]. The systemic impairment of microcirculatory function is a major cause of tissue ischemia and thromboembolic complications seen in severe COVID-19 forms [8]. Based on these two major pathobiological mechanisms involved in the pathogenesis of severe COVID-19, two novel markers have been recently proposed: MR-proADM and the in vitro measurement of IFNγ. Table 3 and Table 4 review the clinical studies investigating the diagnostic and prognostic performance of the two markers in patients with SARS-CoV-2 infection.

MR-proADM has recently emerged as a promising prognostic marker in COVID-19. Several studies, mostly retrospective evaluations of patients during the first pandemic period, have proposed cut-off values ranging from 0.8 to 2 nmol/L to predict adverse outcomes. This heterogeneity likely reflects differences in patient characteristics (age, severity at enrollment, proportion of ICU admittance) and mortality rates between studies (Table 3). An external validation in larger cohorts of patients of the different cut-off values proposed so far could be of value to define which is the optimal threshold of MR-proADM and to identify possible patient- and/or outcome-specific values. Already a few studies compared MR-proADM at admission to other biomarkers (i.e., CRP, PCT, lymphocyte count, LDH, d dimer, NT-pro-BNP) or severity scores (SOFA, APACHE II, SAPS II), finding similar or higher discriminatory power of MR-proADM for mortality or composite outcomes that include disease progression or death [17,20,22,23,24,25,26,27]. Only a small number of studies have so far evaluated MR-proADM longitudinally in COVID-19 patients. Gregoriano and colleagues performed the test on 89 hospitalized patients upon hospital admission (T0) and at days 3/4 (T1), 5/6 (T2) and 7/8 (T3) [19]. Consistently with our findings, MR-proADM values resulted significantly higher in non-survivors compared to survivors at every measured time point. In non-survivors, MR-proADM showed a stepwise increase after baseline, with the highest discrimination between survivors and non-survivors observed at 5–6 days from admission (AUROC at T0: 0.78, AUROC at T2: 0.92).

In our prospective cohort study, we demonstrated a high prognostic performance of MR-proADM for in-hospital mortality both at hospital admission and 7 days after enrolment, with an AUROC of 0.87 (CI95% 0.79–0.94) and 0.91 (CI95% 0.84–0.98), respectively. Compared to other studies, our cohort consisted of a homogeneous population of moderate-to-severe COVID-19 patients, all hospitalized for pneumonia with respiratory failure. Moreover, patients were prospectively enrolled after the first pandemic waves, hence in a time period when hospitals and acute care centers had reacted and reorganized to the pandemic emergency. Thus, it could be a more reliable representation of current and future scenarios, with both patients’ demographic characteristics and ICU admission and mortality rates in line with the current situation [28].

Deregulation of host immunity has been recognized as an important driver of poor prognosis in COVID-19 patients [4,5,6]. Exuberant cytokine production has shown to play an important role in COVID-19 mortality, and immunosuppressive therapy demonstrated to improve clinical outcomes [29,30]. Conversely, the role of IFN production by activated T-cells in SARS-CoV-2 infection remains less clear. From an immunopathological standpoint, IFN-mediated immunity is known to be impaired by COVID-19 itself [6,31]. Yet, clinical data failed to demonstrate a positive impact of systemic IFNβ-1a in SARS-CoV-2 infection [32]. IFNγ-based therapy, while already employed as adjunctive immunotherapy in different bacterial, mycobacterial and fungal infections or sepsis-induced immunodepression, has been largely avoided in COVID-19 due to its potential proinflammatory effects [33,34]. Interestingly, a recent case series of subcutaneous IFNγ treatment for severe COVID-19 pneumonia in five immunocompromised patients with prolonged viral shedding showed beneficial results, with all patients experiencing a rapid decline in SARS-CoV-2 viral loads and four out of five patients demonstrating clinical recovery [35].

In vitro IFNγ production, a novel functional marker of cell-mediated immune response, is being increasingly employed in immunocompromised transplant patients to assess their risk of infection and adverse outcomes [9,10,11]. Only a small number of studies have evaluated its performance in COVID-19 patients so far (Table 4). Compared to healthy controls and subjects with asymptomatic SARS-CoV-2 infection or mild disease severity (i.e., outpatients), hospitalized patients have been characterized by lower in vitro IFNγ production across different cohorts [5,12,36,37]. In a prospective study comparing 28 outpatients to 61 hospitalized COVID-19 patients, Cremoni and colleagues found that IFNγ production inversely correlated with hospitalization risk with an AUROC of 0.92 using 12.1 IU/mL as cut-off (sensitivity 51%, specificity 96%) [12]. Conversely, T-cell exhaustion measured by IFNγ does not seem to differentiate severe COVID-19 from other infective causes of respiratory failure. Indeed, Blot and colleagues found no differences in in vitro IFNγ production at hospital admission between COVID-19 patients and patients with bacterial community-acquired pneumonia (median of 4.42 IU/mL compared to 2.64 IU/mL, *p* > 0.05), whilst the two groups differed by other inflammatory/anti-inflammatory cytokines (GM-CSF, CXCL10, IL-10) [36]. So far, only two studies have evaluated the prognostic role of in vitro IFNγ production in COVID-19 patients [5,37]. No differences were found between survivors and non-survivors in both studies, which in part could be due to the limited number of events. Using the combined outcome of COVID-19-related complications that included deep vein thrombosis, secondary bacterial infections, organ failure, ICU access and death, Ruetsch and colleagues found that patients with IFNγ levels at hospital admission lower than 15 IU/mL were associated with more complications and that this correlation was confirmed even after controlling for lymphocyte count [5].

Our findings on in vitro IFNγ production differ from previous studies for three main reasons. First, we enrolled only patients with moderate-to-severe COVID-19 as defined by respiratory failure and radiologic-confirmed pneumonia. Indeed, median levels of IFNγ production at T0 in our cohort were 4.50 IU/mL, which are comparable to those of patients with the worst prognosis in other cohorts. Second, for all patients, we obtained a follow-up timepoint (T1) 7 days after T0, with median levels further lowered to 4.35 IU/mL. This could suggest that the optimal time to assess cell-mediated immune response in COVID-19 may not be at the beginning of the infection (i.e., “viral replication phase”) but later on (i.e., “host response phase”). This is coherent with the immune alterations that characterize severe COVID-19 during the disease course, with a concomitant increase in proinflammatory cytokines and functional exhaustion of cell-mediated antiviral response [4,5,6]. Third, we analyzed the subgroup of non-vaccinated patients, in whom the prognostic ability of in vitro IFNγ production was shown to be greater. Despite the small sample size, these results led us to hypothesize that the biomarker may be of particular interest in very frail and/or vaccine-unprotected subjects. Further studies are warranted to explore the role of in vitro IFNγ production as a prognostic marker to identify, throughout hospitalization, patients characterized by a higher grade of immunosuppression at risk of adverse outcomes.

The main limitation of our study is the low number of events, which limits the statistical power and statistical model adjustment possibilities. When designing the study, we relied on the available data referring to the first wave of the COVID-19 pandemic, for which both the degree of frailty and the mortality of patients were higher than observed. Yet, rather than choosing surrogate endpoints or combined outcomes, we preferred to evaluate a hard endpoint such as in-hospital mortality. This strengthens our findings, making the results clearer and more comparable to the current situation than those obtained from patients hospitalized in the early phase of COVID-19.

In conclusion, in our prospective cohort of patients with COVID-19 pneumonia and different degrees of respiratory failure, two novel biomarkers associated with coagulopathy and immune system dysregulation showed results worthy of interest in predicting in-hospital mortality. MR-proADM at admission and during hospitalization resulted strongly associated with patients’ death. Low in vitro IFNγ production after one week of hospitalization was associated with mortality in non-vaccinated patients. Although it needs to be confirmed by larger studies, this association suggests a role of in vitro IFNγ production as a possible biomarker for identifying COVID-19 patients with a higher grade of immunosuppression and mortality risk.

## Figures and Tables

**Figure 1 viruses-14-01683-f001:**
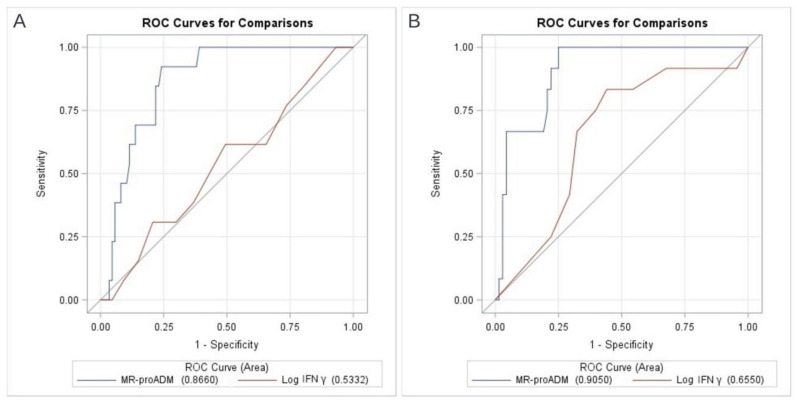
AUROC analysis of MR-proADM and in vitro IFNγ production for in-hospital mortality, at T0 (panel (**A**)) and T1 (panel (**B**)).

**Figure 2 viruses-14-01683-f002:**
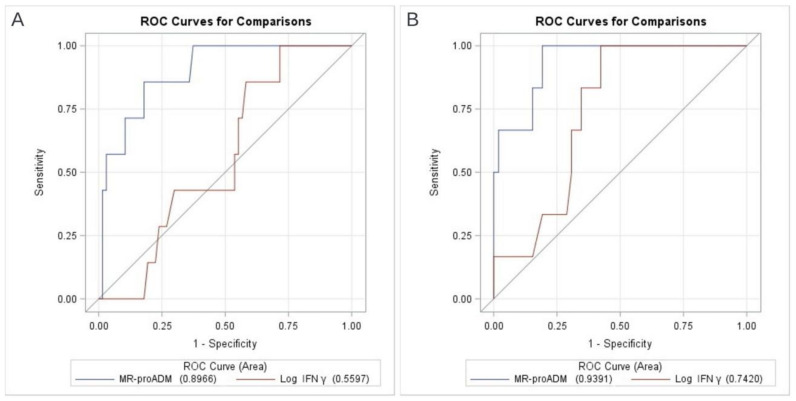
AUROC analysis of MR-proADM and in vitro IFNγ production for in-hospital mortality in the subgroup of non-vaccinated patients, at T0 (panel (**A**)) and T1 (panel (**B**)).

**Table 2 viruses-14-01683-t002:** Association between MR-proADM and in vitro IFNγ production and in-hospital mortality, at admission (T0) and after 7 days of hospitalization (T1).

	Overall	Survivors	Deceased	OR (95% CI), *p* Value #	OR (95% CI), *p* Value §
**T0 (*n* = 100, 13 deceased)**					
MR-proADM, nmol/L	0.84 (0.66–1.20)	0.79 (0.63–1.03)	1.41 (1.12–1.77)	5.03 (1.77–16.32), **<0.001**	3.39 (1.01–11.96), **0.048**
In vitro IFNγ production, IU/mL Log IFNγ production	4.50 (0.85–17.60) 1.50 (−0.16–2.87)	3.90 (0.80–16.80) 1.36 (−0.22–2.82)	5.30 (1.10–20.10) 1.67 (0.10–3.00)	1.04 (0.84–1.27), 0.773	0.86 (0.64–1.12), 0.289
**T1 (*n* = 81, 12 deceased)**					
MR-proADM, nmol/L	0.72 (0.55–1.10)	0.66 (0.53–0.95)	1.67 (1.08–1.96)	9.98 (3.09–39.52), **<0.001**	11.80 (2.73–78.77), **<0.001**
In vitro IFNγ production, IU/mL Log IFNγ production	4.35 (0.75–17.25) 1.47 (−0.29–2.85)	5.80 (0.75–20.95) 1.76 (−0.29–3.04)	1.20 (0.65–3.10) 0.17 (−0.46–1.11)	0.81 (0.60–1.03), 0.083	0.73 (0.49–1.01), 0.057
**Subset of non-vaccinated patients ***					
**T0 (*n* = 74, 7 deceased)**					
MR-proADM, nmol/L	0.82 (0.63–1.05)	0.79 (0.61–0.98)	1.51 (1.12–1.90)	30.25 (4.35–346.36), **<0.001**	N/A
In vitro IFNγ production, IU/mL Log IFNγ production	3.60 (0.90–15.40) 1.28 (−0.11–2.73)	3.50 (0.90–19.10) 1.25 (−0.11–2.95)	4.50 (0.80–5.80) 1.50 (−0.22–1.76)	0.89 (0.62–1.20), 0.536	N/A
**T1 (*n* = 59, 6 deceased)**					
MR-proADM, nmol/L	0.69 (0.54–1.01)	0.64 (0.53–0.88)	1.66 (1.07–1.95)	64.65 (6.77–900), **<0.001**	N/A
In vitro IFNγ production, IU/mL Log IFNγ production	3.85 (0.60–17.90) 1.32 (−0.51–2.82)	6.20 (0.65–20.95) 1.82 (−0.43–3.04)	0.95 (0.50–1.40) −0.05 (.0.69–0.34)	0.60 (0.29–0.96), **0.028**	N/A

Data are reported as median (interquartile range). # *p* value: univariable exact logistic regression model; § *p* value: multivariable exact logistic regression model adjusted by severity score (based on age and PaO2:FiO2 ratio at T0 association with mortality); * excluding vaccinated patients (n° 16) and patients with unknown status (n° 10). *p* values <0.05 are reported in bold. Legend: Log natural (base e) logarithm. N/A: not appropriate due to the low number of events.

**Table 3 viruses-14-01683-t003:** Clinical studies investigating diagnostic and prognostic performance of MR-proADM in patients with SARS-CoV-2 infection.

First Author/DOI	Study Design	Study Period	Population at Enrolment	Mortality Rate	Time for MR-proADM Dosing	Endpoint	MR-proADM Cut-Off Value/Performance
Spoto S. 10.1002/jmv.26676	prospective cohort study	04/2020–06/2020	69 hospitalized COVID-19 patients - 39 (56.5%) admitted to medical ward - 30 (43.5%) admitted to ICU	16/69 (23.2%)	N/A	- 30-day mortality - ARDS	- for mortality prediction: 2 nmol/L - for ARDS development: 3.04 nmol/L
Roedl K. 10.1080/1354750X.2021.1905067	prospective cohort study	03/2020–09/2020	64 COVID-19 ICU patients - 29 (45%) required RRT - 35 (55%) without RRT	17/64 (26.5%)	ICU admission	RRT requirement	1.26 nmol/L AUC 0.685 (95% CI: 0.543–0.828)
Montrucchio G. 10.1371/journal.pone.0246771	prospective cohort study	03/2020–06/2020	57 COVID-19 ICU patients	31/57 (54.4%)	– T0 (≤48 h from ICU admission) – T1 (day 3) – T2 (day 7) – T3 (day 14)	in-hospital mortality	1.8 nmol/L AUC 0.85 (95% CI: 0.78–0.90)
Lo Sasso B. 10.1093/labmed/lmab032	retrospective cohort study	09/2020–10/2020	110 hospitalized COVID-19 patients	14/110 (12.7%)	hospital admission	in-hospital mortality	1.73 nmol/L AUC 0.95 (95% CI: 0.86–0.99, 90% sensitivity and 95% specificity)
Gregoriano C. 10.1515/cclm-2020-1295	prospective cohort study	02/2020–04/2020	89 hospitalized COVID-19 patients	17/89 (19.1%)	– T0 (initial blood draw upon hospital admission) – T1 (day 3/4) – T2 (day 5/6) – T3 (day 7/8)	in-hospital mortality	0.93 nmol/L (at T0) AUC 0.78 (93% sensitivity, 60% specificity and 97% negative predictive value)
Sozio E. 10.1038/s41598-021-84478-1	retrospective cohort study	03/2020–05/2020	111 hospitalized COVID-19 patients	negative outcome (death or orotracheal intubation): 28/111 (25.2%)	hospital admission	negative outcome (death and/or orotracheal intubation)	0.895 nmol/L AUC 0.849 (95% CI: 0.77–0.73, 86% sensitivity and 69% specificity)
Benedetti I. 10.26355/eurrev_202102_24885	prospective observational study	03/2020–04/2020	21 hospitalized COVID-19 patients with ARDS	11/21 (52.4%)	- T0 (admission)- T1 (24 h)- T3 (day 3)- T5 (day 5)	30-day mortality	1.07 nmol/L (at T0) AUC 0.81 (91% sensitivity, 71% specificity)
García de Guadiana-Romualdo L. 10.1111/eci.13511	prospective cohort study	03/2020–04/2020	99 hospitalized COVID-19 patients	14/99 (14.1%)	hospital admission	- 28-day mortality - severe COVID-19 progression (composite of admission to ICU and/or need for mechanical ventilation and/or 28-day mortality)	1.01 nmol/L AUC for 28-day mortality 0.905 (95% CI: 0.829–0.955) and AUC for progression to severe disease 0.829 (95% CI: 0.740–0.897)
van Oers J.A.H. 10.1016/j.jcrc.2021.07.017	prospective cohort study	03/2020–05/2020	105 hospitalized COVID-19 patients with pneumonia	30/105 (28.6%)	hospital admission and daily in the first 7 days	28-day mortality	1.57 nmol/L AUC 0.84 (95% CI: 0.76–0.92)
Girona-Alarcon M. 10.1186/s12879-021-05786-5	prospective cohort study	03/2020–06/2020	20 COVID-19 ICU patients -16 adults with ARDS -4 children with MIS-C	0/20 (0%)	N/A	N/A	N/A
Zaninotto M. 10.1016/j.cca.2021.09.016	retrospective cohort study	11/2020	135 hospitalized COVID-19 patients - Group 1, *n* = 20, MR-proADM ≤ 0.55 nmol/L - Group 2, *n* = 82, MR-proADM > 0.55 nmol/L ≤ 1.50 nmol/L - Group 3, *n* = 33, MR-proADM > 1.50 nmol/L	14/135 (10.4%)	single specimen collection within hospitalization (median time elapsed from hospital admission to MR-proADM measurement = 7 days)	- in-hospital mortality - ICU/sub-ICU admission	N/A
García de Guadiana-Romualdo L. 10.1016/j.ijid.2021.08.058	multicenter prospective cohort study	09/2020–10/2020	359 hospitalized COVID-19 patients	90-day mortality: 32/359 (8.9%)	hospital admission	90-day mortality	0.8 nmol/L AUC 0.832 (95% CI: 0.770–0.894, 96.9% sensitivity, 58.4% specificity and 99.5% negative predictive value)
Mendez R. 10.1136/thoraxjnl-2020-216797	prospective observational study	03/2020–06/2020	210 COVID-19 patients at the ED (23 discharged and managed as outpatients, 179 with initial ward admission, 8 with initial ICU admission). Of these, 97 patients with biomarkers at day 1 and follow-up visit	27/210 (12.8%)	- T1 (ED admission) - T2 (post-hospitalization follow-up visit, median time = 65 days)	in-hospital mortality	1.16 nmol/L
Moore N. 10.1136/jclinpath-2021-207750	prospective cohort study	04/2020–06/2020	135 hospitalized COVID-19 patients	30/135 (22.2%)	hospital admission	30-day all-cause mortality, intubation and ventilation, critical care admission and NIV use	N/A (applied external cut-off values) AUC 0.8441 for 30-day mortality
Minieri M. 10.1186/s13054-021-03834-9	retrospective cohort study	N/A	321 COVID-19 patients at the ED	97/321 (30.2%)	ED admission	in-hospital mortality	1.105 nmol/L AUC 0.85
Oblitas C.M. 10.3390/v13122445	prospective cohort study	08/2020–11/2020	95 COVID-19 ICU patients	12/95 (12.6%)	≤72 h from ICU admission	30-day mortality and combined event (mortality, venous or arterial thrombosis, orotracheal intubation)	1.0 nmol/L AUC for mortality 0.73 (95% CI: 0.63–0.81, positive likelihood ratio and negative likelihood ratio 2.40 and 0.46, respectively), AUC for combined event 0.72 (95% CI: 0.62–0.81, positive likelihood ratio and negative likelihood ratio 3.16 and 0.63, respectively)
Indirli R. 10.1111/eci.13753	retrospective cohort study	03/2020–06/2020	116 hospitalized COVID-19 patients	21/116 (18.1%)	hospital admission	- in-hospital mortality - composite outcome (death, ICU admission, in-hospital complications), length of stay	1.0 nmol/L AUC 0.79 (71.3% sensitivity, 85.7% specificity, 5.0 positive likelihood ratio and 0.33 negative likelihood ratio)

Legend: N/A, not available; HR, hazard ratio; OR, odds ratio; AUC, area under curve; CI, confidence interval; ARDS, acute respiratory distress syndrome; RRT, renal replacement therapy; MIS-C, multisystem inflammatory syndrome; ED, emergency department; CRP, C-reactive protein; PCT, procalcitonin.

**Table 4 viruses-14-01683-t004:** Clinical studies investigating diagnostic and prognostic performance of in vitro IFNγ production in patients with SARS-CoV-2 infection.

First Author /DOI	Study Design	Study Period	Population at Enrolment	Mortality Rate	Time of in vitro IFNγ Production Dosing	Endpoint	In Vitro IFNγ Production Cut-Off Value/Performance
Blot M. 10.1186/s12967-020-02646-9	prospective cohort study	11/2018–02/2020	63 hospitalized patients with severe pneumonia - 27 COVID-19 - 36 non-COVID-19 CAP 7 healthy controls	- COVID-19: 1/27 (3.7%) - non-COVID-19 CAP: 2/36 (5.5%)	≤48 h from hospital admission	30-day mortality	N/A
Ruetsch C. 10.3389/fmed.2020.603961	prospective cohort study	03/2020–04/2020	101 COVID-19 patients - 41 mild disease (outpatients) - 30 moderate disease (medical wards) - 30 severe disease (ICU) 50 healthy controls	6/101 (5.9%)	at baseline (day 0) and follow-up time points up to 2 months after admission to the hospital (not further specified)	disease progression and complications (deep vein thrombosis, secondary bacterial infections, organ failure, ICU access and death)	15 IU/mL
Cremoni M. 10.3389/fmed.2020.608804	prospective cohort study	04/2020–05/2020	29 HCWs with SARS-CoV-2 infection -13 asymptomatic -15 mild disease (outpatients) -1 moderate disease (hospitalized) 60 COVID-19 patients -30 moderate disease -30 severe disease (ICU)	N/A	Blood samples were collected at day 0 of the admission (patients) and at inclusion for HCWs	hospitalization	12.1 IU/mL AUC 0.92 (51% sensitivity, 96% specificity)
Dhanda A.D. 10.1016/j.imbio.2022.152185	prospective cohort study	04/2020–05/2020, 02/2021	41 hospitalized COVID-19 patients - 11 with oxygen support - 1 in ICU at admission 12 healthy controls	7/41 (17.1%)	at baseline	in-hospital mortality	N/A

Legend: N/A, not available; AUC, area under curve; CAP, community-acquired pneumonia; HCWs, health care workers; CRP, C-reactive protein.

## Data Availability

The data presented in this study are available on request from the corresponding author.

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
