# Peer review of "Prognostic Value of Mid-Region Proadrenomedullin and In Vitro Interferon Gamma Production for In-Hospital Mortality in Patients with COVID-19 Pneumonia and Respiratory Failure: An Observational Prospective Study"

_viruses, 2022, doi:10.3390/v14081683_

Round 1

Reviewer 1 Report

The authors present a very well prepared and well written manuscript describing the prognostic value of MR-proADM and in vitro interferon gamma production in COVID-19 patients and how these biomarkers could potentially predict clinical outcomes and survival. Although the MR-proADM provides a clear correlation with clinical outcome, in vitro IFNg levels were less clear as a predictor of adverse outcome making the manuscript less appealing. The study of MR-proADM as a prognostic marker has been studied extensively and the current study further validates these findings.

1. The use of the in vitro IFNg levels as a biomarker is less clear and the authors take some liberties with non-significant findings and suggest that this may also be important. This could be accounted for by low numbers of samples or due to random chance since these did not reach statistical significance. I would recommend that the authors change their wording in places where they state that differences occurred. Example (Abstract; lines 31 and 32)

2. There was very little discussion about the role of T-cells in the COVID-19 samples tested in vitro for IFNg production. The authors use a standardized protocol for specifically stimulating T-cells and then follow up with measurement of IFNg however there is  no discussion relevant to the impact of stimulating T-cells in this manner that would provide some mechanistic insight regarding the IFNg changes.  I think additional discussion regarding the role of T-cells at T0 and at T1 and how IFNg production might be effected at these timepoints would improve the manuscript and provide broader appeal.    

Author Response

see cover letter (attached)

Reviewer 2 Report

This manuscript described interesting theme and entire structure was well arranged. Relatively small number of deceased patients may attenuate detection power. However, the usefulness of MR-proADM for prognostic estimation in hospitalized COVID-19 patients was well addressed as in previous studies. What is the novelty of your study compared with previous studies? Additionally, I want to know the relationship between MR-proADM and other clinical markers like CRP, lymphocyte or D-dimmer.

Author Response

see cover letter (attached)
